# A Systematic Review for Vaccine-Preventable Diseases on Ships: Evidence for Cross-Border Transmission and for Pre-Employment Immunization Need

**DOI:** 10.3390/ijerph16152713

**Published:** 2019-07-30

**Authors:** Varvara A. Mouchtouri, Hannah C. Lewis, Christos Hadjichristodoulou

**Affiliations:** 1Department of Hygiene and Epidemiology, Faculty of Medicine, University of Thessaly, 41222 Larissa, Greece; 2Department of Infectious Disease Epidemiology, Robert Koch Institute, 13353 Berlin, Germany

**Keywords:** vaccine, cruise, ship, travel, maritime health, varicella, chickenpox, mumps, rubella, measles, pertussis, diphtheria, meningococcal disease, hepatitis A, vaccination, occupational health, seafarers, sailors

## Abstract

A literature review was conducted to identify evidence of cases and outbreaks of vaccine-preventable diseases (VPDs) that have been reported from on board ships and the methods applied on board for prevention and control, worldwide, in 1990 to April 2019. Moreover, evidence from seroprevalence studies for the same diseases were also included. The literature review was conducted according to Preferred Reporting Items for Systematic reviews (PRISMA) guidelines. A total of 1795 cases (115 outbreaks, 7 case reports) were identified, the majority were among crew (1466/1795, 81.7%) and were varicella cases (1497, 83.4%). The origin of crew cases was from sub-tropical countries in many reports. Measles (40 cases, 69% among crew), rubella (47, 88.7%), herpes zoster (9, 69.2%) and varicella cases (1316, 87.9%) were more frequent among crew. Mumps cases were equal among passengers and crew (22/22). Hepatitis A (73/92, 70.3%), meningococcal meningitis (16/29, 44.8%), and pertussis (9/9) were more frequent among passengers. Two outbreaks resulted in 262 secondary measles cases on land. Review results were used to draft a new chapter for prevention and control of VPDs in the European Manual for Hygiene Standards and Communicable Disease Surveillance on Passenger Ships. Despite past and current evidence for cross-border VPD transmission and maritime occupational risks, documented pre-employment examination of immune status, vaccination of seafarers, and travel advice to passengers are not yet regulated.

## 1. Introduction

Ships are long acknowledged as semi-closed and densely populated environments with close living and sleeping quarters, and shared water, ventilation and sewage systems [1,2,3]. Such environments are accountable to a constant flux of people from over the world and conducive for the spread of communicable diseases.

Approximately 397 million passengers embarked and disembarked in European Union (EU) ports in 2016 [4]. It was estimated that 26.6 million tourists spent holidays on cruise ships worldwide in 2017 [5], while a total of 1,647,500 seafarers work on merchant ships operating internationally over the world [6]. The risk of cases and outbreaks of disease among the population on board ships is ever present.

Outbreaks of norovirus and influenza, as well as food and waterborne disease, are well documented [7,8,9,10]. In recent years, outbreaks of diseases, which could be prevented by routine vaccines (e.g., measles, rubella, and varicella) have also been reported on ships [11,12]. This is not surprising as both crew and passengers originate from diverse countries with variable vaccine schedules and coverage. However, even in countries where measles was declared eliminated, land-based community outbreaks are currently ongoing [13]. In response to these events, governments and international agencies have developed national [14] and international guidelines [15,16] to provide advice for timely prevention and control.

In light of events constituting cross-border health threats under the legal framework of the International Health Regulations (2005) and the Decision no 1082/2013/EU on serious cross-border threats to health, updated European guidance was also requested by EU Member States [17,18]. As such, the EU SHIPSAN ACT joint action [19] chose to update the “European Manual for Hygiene Standards and Communicable Disease Surveillance on Passenger Ships” to include a chapter on vaccine-preventable diseases (VPDs) [16]. In writing up-to-date guidelines on VPDs, the evidence-base for the occurrence of disease and characteristics of outbreaks or other events that could be prevented by routine vaccination, on ships, and the methods applied for their prevention and control, was reviewed. To our knowledge there has been no other review published with this objective. 

## 2. Materials and Methods

### 2.1. Research Question and Objectives

The bibliographic review was conducted in order to answer the following research questions:What is the published evidence of cases and outbreaks of measles, mumps, rubella, varicella, diphtheria, tetanus, pertussis, meningococcal disease, hepatitis A, and hepatitis B that have been reported from on board ships and the methods applied on board for the prevention and control, worldwide, since 1990?What is the published evidence of exposure of travelers to measles, mumps, rubella, varicella, diphtheria, tetanus, pertussis, meningococcal disease, hepatitis A, and hepatitis B, based on serological examinations, worldwide, since 1990?

This review aims at giving an insight on VPDs on board ships that have occurred worldwide, by analyzing evidence published since 1990. Results of this review were used in the revision of the “European Manual for Hygiene Standards and Communicable Disease Surveillance on Passenger Ships” [16].

The specific objectives of the review were the following:To undertake descriptive epidemiology (person, place and time characteristics) of cases and outbreaks of VPDs (measles, mumps, rubella, varicella, diphtheria, tetanus, pertussis, meningococcal disease, hepatitis A, and hepatitis B) that have been reported from on board ships since 1990.To describe the cause/risk factors identified for introduction and transmission of these VPDs onto ships.To describe key methods applied for the prevention and control of these VPDs on ships.

### 2.2. Search Strategy

The search concepts used for the above-mentioned topic are: (a) Public health event: Cases or outbreaks, or evidence for exposure based on serological examinations to VPDs including measles, mumps, rubella, varicella, herpes zoster, diphtheria, tetanus, pertussis, meningococcal disease, hepatitis A, and hepatitis B; (b) type of intervention: Prevention and control measures of those VPD; (c) population of interest: Humans travelling with ships; (d) setting: Ships; (e) outcome: Effectiveness of measures applied, cost, public health impact. 

The systematic review was performed according to the procedures and checklist outlined by the Preferred Reporting Items for Systematic reviews (PRISMA) [20]. To retrieve information, Pubmed was searched for relevant articles published between 1 January 1990 and 30 April 2019 in English, German or Greek. Moreover, the German Institute of Medical Documentation and Information (DIMDI) databases (includes Medline, Embase, Embase Alert, Biosis Previews, SciSearch) were searched for relevant articles published between 1 January 1990 and 15 July 2015 in any of the official EU languages. All text fields were searched using the following search terms:

(ship OR ferry* OR boat* OR yacht* OR cruise* OR barge* OR “fishing vessel” OR tanker* OR “on board” OR aboard OR “maritime transport”) AND (chickenpox OR diphtheria OR “German measles” OR hepatitis A OR hepatitis B OR measles OR “meningococcal disease” OR meningitis OR mumps OR pertussis OR rubella OR tetanus OR varicella). An asterix was used for abridged terms. Within articles selected for full-text analysis, the reference lists were checked for completeness.

The SHIPSAN TRAINET communication platform was searched for relevant vaccine-preventable disease cases or outbreaks between 29 March of 2011 (establishment of platform) and 30 April 2019 [21]. Furthermore, the World Health Organization, the European Centre for Disease Prevention and Control (ECDC) and the United States Centers for Disease Control and Prevention (US-CDC) Division of Global Migration and Quarantine (DGMQ) were contacted separately for surveillance or outbreak reports relating to VPDs outbreaks on ships. 

### 2.3. Inclusion and Exclusion Criteria

Inclusion criteria were: Articles or reports or record reviews or other documents published in peer-reviewed journals or national and international organizations’ publications or information through personal communications, from January 1990 until April 2019, that reported any seroprevalence evidence or clinically and/or laboratory confirmed case or outbreak of measles, mumps, rubella, varicella, diphtheria, tetanus, pertussis, meningococcal disease, hepatitis A, and hepatitis B, on board ships sailing worldwide. 

Exclusion criteria were: Publications reporting influenza were excluded from the review since there was existing recommended policy in the “European Manual for Hygiene Standards and Communicable Disease Surveillance on Passenger Ships” [16]. Moreover, those VPDs where there was not considered a risk specifically related to ships (e.g., Japanese encephalitis, rabies) and for risk groups based on age, occupation, underlying health condition, etc. (e.g., pneumococcal disease, rotavirus) were also excluded. Articles for which the full text was not available in English or German or Greek were also excluded. Moreover, full-text reports/articles were excluded if: no cases of VPDs reported (including review articles with no original data), data outside time range, different versions of same report/article, reports from hospital ships, or ships used to house displaced populations/refugees.

### 2.4. Data Extraction and Analysis

The quality of articles included in the review were assessed on the basis of completing the inclusion criteria. Specific questions and a data extraction sheet were used by the two researchers to independently and systematically review and extract the data from the publications. The data extraction sheet was pilot-tested on records/articles and then refined accordingly.

Duplicates and reports including data from prior to 1990 were excluded (Figure 1). Only references for which full text could be obtained and; therefore, analysis performed were included. For events with multiple reports, the original report was included and additional reports were only included if they contained additional information relevant for data extraction. For all included articles, full-text analysis was performed.

We contacted authors from one paper for further information: Nieto Vera et al. Measles outbreak in Campo de Gibraltar, Cadiz, Spain, during the period February–July 2008 [22]. The number of laboratory-confirmed, epidemiologically-linked, and clinically compatible cases was clarified. As was the onset date of the last case and the number of hospitalizations. 

### 2.5. Ethical Approval

This review concerns a review of already published material, and; therefore, ethical approval was not required. Moreover, data extracted from the EU SHIPSAN information system (https://sis.shipsan.eu/) owned by the EU SHIPSAN ACT partnership who approved the use of data for the systematic review and had the tasks to conduct this literature review and to revise the “European Manual for Hygiene Standards and Communicable Disease Surveillance on Passenger Ships” in the framework of the EU SHIPSAN ACT joint action [16].

## 3. Results

### 3.1. Reports Included in the Review

Twenty-six articles/reports fulfilled the eligibility criteria (Figure 1). Additionally, eight records in the EU SHIPSAN ACT joint action information system and one personal communication reported cases or outbreaks of VPDs associated with ships were included in this review (Figure 1) [21,23]. The systematic review revealed 24 events including 17 outbreaks and seven single case events, which occurred between 1 January 1990 and 30 April 2019 on passenger, cargo, military, and work ships (Table 1, Table 2 and Table 3). Moreover, the systematic review analyzed data from five studies that presented results from review of ships’ or of authorities’ records and four studies reporting serological test results of seafarers.

In total, 1795 cases (115 outbreaks, seven case reports) were identified, the majority were among crew (1466/1795, 81.7%) and were varicella cases (1497, 83.4%). The origin of crew cases was from sub-tropical countries in many reports. Measles (40 cases, 69% among crew), rubella (47, 88.7% crew), herpes zoster (9, 69.2% crew), and varicella cases (1316, 87.9% crew) were more frequent among crew. Mumps cases were equal among passengers and crew (22/22). Hepatitis A (73/92, 70.3%), meningococcal meningitis (16/29, 44.8%), and pertussis (9/9) were more frequent among passengers.

### 3.2. Single Case Reports

There were single case reports of each of diphtheria (in 1997), hepatitis A (in 2011), meningococcal meningitis case (in 2017), and varicella (in 2014 and 2018) from passenger ships, a meningococcal meningitis case from a military ship (in 2003) and varicella (in 2015) from cargo ships (Table 1) [21,24,25,26,27]. Six of the seven case reports were in crew with the exception being the diphtheria case, which was in a passenger [24].

Except from one death reported due to varicella [25], there were no other deaths reported. Τhe case of meningitis (in 2017) and the diphtheria case were the only cases reported to be hospitalized ashore [21,24].

Regarding source of infection for the diphtheria case reported in a passenger, characterization of the diphtheria isolate found that the strain was indistinguishable from the predominant epidemic strain that was currently circulating in the countries of the former Union of Soviet Socialist Republics, providing evidence that the infection was likely acquired during the Baltic cruise [24]. 

Response measures were described in six out of the seven reports. In the three cases of varicella, isolation, case finding/active surveillance in crew and crew vaccination were applied and cases were reported to authorities [21,25]. Treatment, cleaning/disinfection and risk communication were described to be applied in response to one varicella case [21]. In response to the diphtheria case, diphtheria antitoxin and antibiotics were given to each case and close family, while contacts received antibiotic prophylaxis and low dose diphtheria vaccine boosters. Meningitis cases were isolated, in one case chemoprophylaxis was given to close contacts from the ship and contact tracing conducted [21,26].

### 3.3. Outbreaks Reported (Events with More Than One Case Involved)

#### 3.3.1. Characteristics of Outbreak Setting and Affected Population

The review revealed 17 outbreaks of VPDs that have occurred world-wide based on the reports that fulfilled the eligibility criteria, between 1996 and 2019. Those outbreaks were varicella (*n* = 5), measles (*n* = 3), rubella (*n* = 3), a multi-pathogen varicella–measles–rubella outbreak (*n* = 1), hepatitis A (*n* = 3), meningitis (*n* = 1), and mumps (*n* = 1). Outbreaks were associated with passenger ships (13 cruise ships and one ferry), military (navy) ships (*n* = 2), and a cargo ship (*n* = 1) sailing world-wide (Table 1). Most cases were reported on ships associated with hepatitis A, varicella, rubella, and measles outbreaks, with 71, 63, 43, and 37 cases, respectively (Table 4). Two articles reported 262 secondary measles cases on land [22,28].

The duration of the majority of the outbreaks (*n* = 13) were protracted (over one month long) (Table 1) with the longest duration being for a varicella outbreak that spread between crew members of different ships (10 months) [23] and a measles outbreak that spread onto land (six months) [22,28]. Only one death was reported, during the meningitis 2012 outbreak (case fatality rate 25%) [29]. Hospitalizations were reported in seven outbreaks (7/17, 41%) and the hospitalization rate varied between 8% to 100%, being highest in the meningitis 2012 (*n* = 4/4, 100%) and hepatitis A (*n* = 20/34, 59%) outbreaks (Table 1) [29,30].

The majority of outbreaks involved crew members (14/17, 82%). Eleven outbreaks (11/17, 65%) affected only crew members and in two further measles outbreaks [22,28,31], crew were identified as the index cases leading to secondary crew cases, cases in passengers, and those on land [22,28]. One varicella outbreak in 2012 affected both crew and passengers and the index case was not reported [21]. In the three outbreaks of hepatitis A, all cases were reported in passengers on Nile cruises. 

The total number of crew affected in outbreaks (*n* = 145) were almost double the number of passengers cases (*n* = 78) (Τable 4). Crew attack rates, calculated using total crew as the denominator, ranged from <1% to 6% (Table 1). It should be noted that the two highest attack rates (4% and 6%) were derived from sero-surveys after confirmed rubella outbreaks, which included asymptomatic cases (approx. 50% of cases in both studies); the highest attack (6%) was for a rubella outbreak on a Navy ship where no control measures were documented and the outbreak only ended after the ship’s return [3]. The mumps outbreak had a crew attack rate of 4%, and a crew attack rate of 2.4% was calculated from the data available for the measles outbreak in 2014 [31]. Insufficient data were available for most varicella outbreaks to calculate the attack rate, but in two outbreaks it was less than 1% of crew (Table 1) [1,32]. In those outbreaks and events, which documented the nationality of crew from passenger ships (*n* = 10) (Table 1), all noted crew from a range of countries (some ships had crew from over 130 countries), including a substantial proportion of crew originating from sub-tropical/tropical countries (Table 4). Insufficient data were reported to describe attack rates, the risk of secondary transmission to, or nationality of affected passengers.

#### 3.3.2. Secondary Cases on Land

Two measles outbreaks linked to passenger ships are of particular interest as they reported a large number of cases and exemplify the risk of transmission of VPDs among crew, passengers and to others on land [22,28]. A measles outbreak in 2014, which started on a cruise ship in the western Mediterranean in 2014 [31], and another related to a fast ferry operating on the Algeciras (Spain)-Tangier (Morocco) route in 2008 [22], both led to extended outbreaks involving a substantial number of cases on land in Italy (*n* = 110) [28,31] and Spain (*n* = 152), respectively [22]. For the ferry outbreak in 2008, there were three index cases, two of which were crew members, which disembarked the ferry and led to a large outbreak in the county of Campo de Gibraltar in Spain [22]. In total 155 cases of measles were reported in the county over six months between February and July 2008, of which 88% (*n* = 137) were laboratory confirmed. An incidence rate of 112.3 cases of measles per 105 population was reported in the Province of Algeciras which experienced most (83%) of the cases.

For the 2014 measles outbreak on a cruise ship, as of 10 March 2014, there were 29 laboratory confirmed cases, 80% (*n* = 23) of the cases were crew [3,31]. Twenty-three of the 968 crew (2.4%) and six of approximately 3352 passengers (0.2%) became infected with measles over a three-week period. In addition to sporadic contacts of primary cases becoming infected, two secondary outbreaks linked to passenger cases and involving extensive nosocomial transmission were reported on land [22,28]. One secondary outbreak occurred in Brindisi province, Puglia Region and involved 32 cases to 8 May 2014 [33]. The outbreak was spread within an emergency department of a hospital (including two health care worker cases) and then in the community to the fifth generation. The other secondary outbreak was even more extensive and led to 80 cases to July 2014 in Sardinia with transmission in families, work places and the hospital setting [28]. Nosocomial cases included health care workers (*n* = 15, 19%) and patients infected while in the emergency department or hospital ward (*n* = 29, 36%). In total, 45% (35/78) case-patients were hospitalized. In all these secondary outbreaks, most cases were young adults, but children including those <2 years old were also affected (Table 1).

#### 3.3.3. Vaccination History of Travelers

Half of the outbreaks involving crew (7/14, 50%) documented the vaccination history or susceptibility rate of crew members and all found insufficient vaccination and/or documentation (Table 1). In all three rubella outbreaks, a substantial proportion of crew were found to have negative or no documentation of rubella vaccination or immunity (75%, 81%, and 96%, respectively) [3,34]. In the measles 2014 outbreak, 85% of all crew and 21/24 cases (87.5%) had a negative or unknown vaccination history [28]. In the measles 2008 outbreak, 72% of cases had not been previously vaccinated [22]. In the mixed rash 2006 outbreak, only three crew members (<1%) had proof of immunity (a vaccine record) to measles or rubella [1]. Three of nine (33%) mumps cases had a mumps or measles–mumps–rubella (MMR) vaccine documented in the mumps 1992 outbreak [35]. For the three hepatitis A outbreaks involving only passengers, all reports documented that none of the cases had been vaccinated against hepatitis A [30,36,37]. 

#### 3.3.4. Source of Outbreaks

Although most reports could determine the index case or cases of outbreaks, little data exists on the source of outbreaks or risk factors for transmission. The source of the three likely inter-linked hepatitis A outbreaks affecting only passengers was hypothesized to be a continuing common source linked to Nile river cruises [30,36,37].

For the 12 outbreaks for which a crew member(s) was identified as the index case, only four outbreak reports discussed the likely exposure/source and believed the source of the outbreaks to have been exposure of crew on land, either before deployment or at one of the stops (Table 1).

In the rubella 1996 outbreak, three primary crew cases, all Germans, were thought to be infected off-ship and pre-deployment, as there became sick 8 to 10 days into deployment (within the incubation period), it could not be determined if the cases had the same source [3]. Sleeping in the ratings deck (basic rank, slept 40–80 per dorm and shared air-conditioning) was determined to be a risk factor for becoming infected with rubella in this outbreak, with sleeping conditions likely favoring transmission. 

In the measles 2014 outbreak, a common exposure on board could not be found and the index crew member (origin not stated; 71% of crew from Asia) was thought likely infected at one of stops [31] or on board [28].

In the mixed rash outbreak in 2006, it was concluded that rubella was imported from the Philippines (Filipino crew member boarded nine days prior to onset), measles from the Ukraine (Ukrainian crew member boarded 13 days prior to onset) and varicella from unknown source country (possibly one of the ship stops as crew members boarded 22 days prior to onset) [1]. The authors also commented that close crew interactions outside the work environment played a role in the spread of the outbreak as there was sustained varicella transmission among crew members with different occupations.

In the meningitis 2012 outbreak, all cases were working in the ship’s kitchen and originated from different countries in three different continents [29].

In two reports involving rubella, the potential risk of congenital rubella syndrome for passengers or crew of childbearing age was estimated. In one report, 33% of passenger responders were of childbearing age and 0.8% were pregnant [34]. In the other report, three crew members and 0.8% of passengers were pregnant [1].

#### 3.3.5. Response Measures

All outbreak reports included reference to at least one response measure being used to control the outbreak (Table 1). In over half of the outbreaks, a report was sent to competent authorities (11/17, 65%) and isolation (10/17, 59%) were applied. Active surveillance or case finding in crew (8/17, 47%), risk communication with crew and/or passengers (8/17, 47%), and treatment of cases (7/17, 41%) were also commonly cited. Contact tracing (4/17, 23.5%), case finding/active surveillance in passengers or others (5/17, 29%), and cleaning/disinfection (4/17, 23.5%) were less frequently cited control measures. Only in one outbreak (measles 2014) was quarantine implemented and in two outbreaks (measles 2014 and mumps 1992) was disembarkation applied (Table 1) [3,31,35]. Vaccination was described as a post-prophylactic measure for crew in ten measles, varicella, and/or rubella outbreaks; and for passengers in two measles outbreaks. The number of vaccines administered were cited in four outbreaks: a total of 127/1020 (12.5%) crew received vaccine in the measles 2011 outbreak which involved five crew cases [23], more than 400 crew on five different ships (total number of crew not stated) in the varicella 2008 outbreak that included 28 crew cases [23], 865/900 (96%) crew in the rubella April to June 1997 outbreak with seven crew cases [34], and 1191/1200 (99%) crew in the mixed rash outbreak 2006 with 16 crew cases [1]. No outbreak report described the use of immunoglobulin for post-exposure prophylaxis in either crew or passengers. However, it was described as used in close contacts on land in the extended measles outbreak in 2008 [22]. Costs of control measures to interrupt transmission were only cited in the mixed rash outbreak in 2006 that involved 16 crew cases on a passenger ship and was estimated at 67,000 USD for vaccinations, supplies, and health department staff time [1]. Specific guidelines were not cited as being applied or being recommended apart for in the measles 2014 outbreak, where in the absence of contact tracing guidelines specifically for ships, the Risk Assessment Guidance for Infectious Diseases transmitted on Aircraft (RAGIDA) guidelines for aircraft were recommend to be adapted [28]. 

### 3.4. Events of VPDs Identified through Record Review Studies

Record reviews identified a total of 1572 cases (1321 crew and 251 passengers). The four record reviews of surveillance data were undertaken between 2005 and 2015, for periods of six months, two years, and five years, the latter being for two reviews (Table 4). One review was of cruise ship medical logbooks from 34 cruise ships sailing into north America, 2009–2010 [38], one of cruise and cargo ship reports to Hamburg Port Health, Germany from world-wide cruises, November 2007 to April 2008 [39] and two of the US Centers for Disease Control and Prevention national database of cruise ships sailing in US waters, 2005–2009 [40,41]. 

These record reviews also found that the majority of cases reported were crew (78–100% of cases); one US study found that only crew were responsible for being the index cases of outbreaks of varicella [38]. However, passenger cases and their contacts were reported in addition to crew members in two reviews (36 and 79 cases, respectively) (Table 4). Both US studies also found that a substantial proportion of varicella cases are associated with outbreaks on-board: One review found that 70% of crew cases (*n* = 66) were accountable to 18 clusters/outbreaks (defined as two or more epi-linked cases, two to nine cases per outbreak) and the other review found that 47.6% of crew and passenger cases (*n* = 89) were accountable to eight outbreaks (defined as five or more epi-linked cases, six to 26 cases per outbreak) [38,40].

The first study found that the majority of cases were first-generation (68%); however, there were also second and third-generation cases reported [40]. Two varicella record reviews recorded the age and sex of infected crew and found the majority to be male (100% and 80%, respectively) and the age range was 26–42 years [39] and 20–66 years (median 29 years) [40], respectively. 

Only the German record review conducted in four ships addressed the possible source of the cases and identified non-immune employers as having been infected from an infected child (*n* = 1) or other crew members on board (*n* = 3) [39]. The US studies described control measures in-line with the published US-CDC Guidance’ Guidance for Cruise Ships on Varicella (Chickenpox) Management’ [14] and the German study in-line with the Port Health Centre recommendations (listed in the paper, no guideline referenced). Both recommendations included isolation of cases and contact tracing to allow for contact (post-exposure) vaccination. According to the authors, such recommendations were followed and were useful in controlling the outbreaks.

Records of reported deaths in the US revealed five deaths due to meningitis/meningococcal disease, three of them were among passengers [11].

### 3.5. Evidence of VPDs Based on Serological Tests on Seafarers

Three seroprevalence studies were undertaken in ships’ crew for hepatitis A, B, and C in the 1990s and one for varicella in 2008 (Table 3). In addition, two of the rubella outbreak reports [3,34] and one varicella outbreak report [32] included seroprevalence studies. These studies found that 11% of cruise ship crew and 12% of navy ship crew were acutely infected or susceptible to rubella; and 13% [32] and 16.5% [42] of crew were susceptible to varicella in two different studies, respectively. For the varicella outbreak it was estimated that “the ship population was two to three times more susceptible to varicella than an age-comparable US population” [32]. Hepatitis A susceptibility of ship crew was found to be between 85% to 100% and was positively associated with age in two studies (Table 3) [43,44,45]. Anti-hepatitis A virus (HAV) and anti-hepatitis B virus (HBV) positivity was found associated with those who had international deployment [43,44].

In the varicella 2008 study, the cost of antibody testing and vaccination was 7000 USD and the paper concludes that testing for antibodies followed by vaccination is a cost-effective method to prevent outbreaks on cruise ships and a recommended mandatory part of the pre-employment medical examination for Indian seafarers [42].

## 4. Discussion

The literature review showed that despite past and current evidence for VPDS cases and outbreaks on ships, cross-border VPDs transmission and maritime occupational risks, documented pre-employment examination of immune status, vaccination of seafarers, and travel advice to passenger are not routinely conducted.

Most outbreaks and cases of VPDs identified were among crew members of cruise ships and were varicella cases. Furthermore, the majority of outbreaks identified a crew member as the index case and the origin of crew cases was from sub-tropical countries in many reports. Immune status of crew members varies depending on the country of origin. Crew originating from countries with low immunization rates and/or ongoing disease transmission could be index cases of outbreaks on board ships [46]. Few data were identified in the literature about crew susceptibility. Seroprevalence studies or surveys as part of outbreak response demonstrated approximately 13% to 16% susceptibility of cruise ships’ crew to varicella [32,42], 11% to rubella [3,34], and high susceptibility of merchant seamen to HAV (85%) [43]. VPDs such as varicella and measles can cause serious complications to adults and an outbreak could affect the normal ship operation especially on ships with a small number of crew sailing on long voyages [47]. In cruise ships, crew members could potentially expose large number of passengers to pathogens depending on their duties and position such as staff in medical facilities and beauty salons, waiters, bartenders, receptionists, and food handlers.

The International Medical Guide for Ships (IMGS) recommends immunization of susceptible crew members after diagnosis of the first case of varicella on board a ship, as a measure to prevent an outbreak, but not as a routine precautionary measure. In the US, varicella is the most frequent VPDs reported on ships and one death of varicella pneumonia was reported in 2015 [11,25]. As a regular pre-employment condition, IMGS requires immunization for diphtheria and tetanus for every seagoing person, while for hepatitis A and hepatitis B for any crew member assigned to medical care duties [47]. For all other crew, IMGS suggests immunization against hepatitis A and hepatitis B as a “wise precaution”. Vaccinations for diseases including cholera, influenza, Japanese encephalitis, meningococcal disease, poliomyelitis, rabies, typhoid fever, and yellow fever would depend on ship destinations and type of freight carried according to IMGS [47]. According to the WHO (World Health Organization) Handbook for Inspection of Ships and Issuance of Ship Sanitation Certificates, inspectors when inspecting ships to issue Ship Sanitation Certificates should ensure that a list of crew members taking care of children is available, indicating the vaccines they have received [48].The Maritime Labour Convention (MLC) of the International Labour Organization (ILO) provides that competent authorities adopt regulations to prevent diseases of seafarers on board [49]. Such regulations can include specific vaccinations and assessment of immune status as a condition for employment of seafarers.

The literature review provided evidence for outbreaks of varicella, measles, rubella, mumps, hepatitis A and meningococcal meningitis in crew. Cruise ships should gain adequate proof of immunity from crew members and provide vaccination to those with inadequate proof, after considering contraindications. In addition to the recommended vaccinations in the IMGS, we recommend vaccination for measles, mumps, varicella, and rubella to be considered in the pre-employment policy of seafarers, rather than as a response measure to prevent outbreaks after the first case occurred. Outbreak control measures and medical evacuations at sea are very resource intensive [1,38,46], so prevention policies involving vaccinations would be more cost effective than control. 

The literature review revealed outbreaks of measles, rubella, herpes zoster, varicella, mumps, hepatitis A, meningococcal meningitis, and pertussis among passengers of cruise ships. Passengers need to be advised about vaccinations before travel, including for measles, mumps, varicella, and rubella [31,34,40,42]. Especially travelers of at-risk groups should be vaccinated according to instructions from family doctors and in sufficient time to develop immunity before boarding ships. Before travelling, pregnant women should check for immunity for infectious disease and update immunizations as needed [50]. Two authors, during investigations of rubella and varicella outbreaks, found pregnant crew members and passengers among the potentially exposed on board ships [1,34]. Vaccination of crew can protect pregnant passengers and crew from exposure to VPDs that can affect pregnancy.

Global data on measles, which has been characterized as the most contagious disease, demonstrate a significant increase in the last two years and this trend continues into the first quarter of 2019, according to WHO [51]. Based on reports received by 30 EU/EEA Member States, ECDC has reported that there is an ongoing large measles outbreak involving more than 44 thousand cases in the last three years [52]. This global increase in measles outbreaks could affect maritime transport since index cases are exposed ashore [3,22,31]. Outbreaks on ships can result in cross-border disease transmission, as reported in two measles outbreaks where secondary cases occurred on land in 2014 [3,31,33]. Preparedness for response, as well as prevention strategies, are advised to be adopted by shipping companies. In case of hospitalizations and laboratory diagnosis conducted ashore, communication of public health information from ship doctors to port health authorities and health care services on land and vice-versa are important to effectively and timely apply control measures [46]. The European Commission new provisional edition of the document for a Regulation on the European Maritime Single Window, suggest the establishment of a ship sanitation database for storing Maritime Declaration of Health, which is expected to facilitate risk assessment and enhance communication between port health authorities and ships [53]. 

The objectives of this review were achieved by collecting evidence mainly from published data. It is possible that most cases and outbreaks are not published in the scientific literature. The majority of cases were varicella cases collected by the US reporting system. Data collected in this review are mainly from US and Europe. In Europe, even if there is an information system available to MS to record cases and public health measures information, it is not systematically used in all EU MS. Moreover, it is possible that cases or outbreaks have been reported to other non-EU countries, but those were not considered in our study. It is unknown how many cases and outbreaks are not reported to competent authorities. 

The consortium of the EU SHIPSAN ACT joint action conducted the current literature review in order to use its results to create a chapter for the prevention and control of VPDs in the European Manual for Communicable Disease Surveillance and Hygiene Standards on Passenger Ships [16]. This manual provides guidance to shipping companies and to port health authorities about pre-embarkation and everyday preventive measures, as well as in response to cases and outbreaks. It suggests that seafarers should maintain written proof of vaccination to demonstrate when needed during pre-employment medical examinations and during voyages. Shipping companies should perform checks of immune status when needed, conduct occupational risk assessments, administer vaccinations for varicella, measles, and rubella in agreement with seafarers and keep up-to-date records of their staff [16]. Passengers should seek travel advice before cruising. Surveillance for early detection of VPD cases and isolation of cases, contact tracing and reporting to competent authorities are also important measures. Standard operating procedures should be available to port health authorities’ contingency plans on how to respond to events of VPDs. Shipping companies’ prevention and control policies should also include standard operating procedures about response measures to VPD events.

## Figures and Tables

**Figure 1 ijerph-16-02713-f001:**
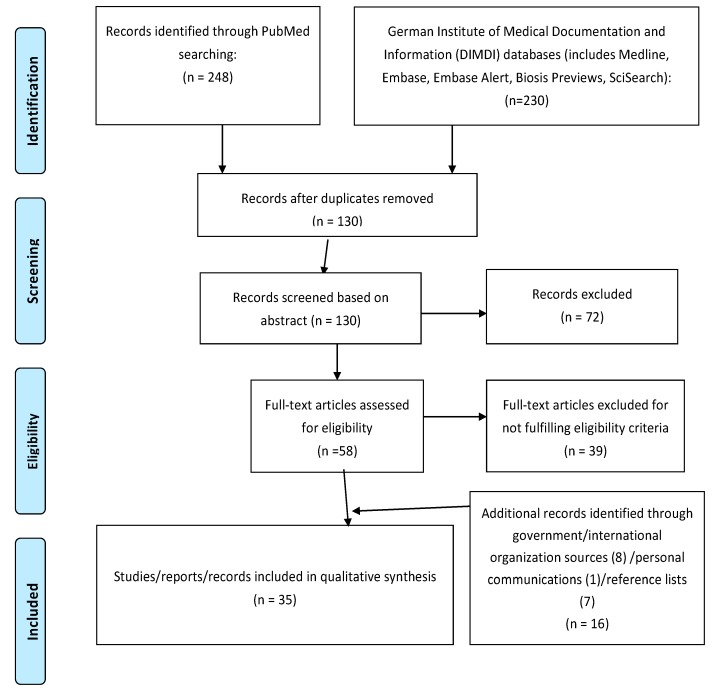
Flow chart of studies included in the review.

**Table 1 ijerph-16-02713-t001:** Descriptive analysis of outbreak and case reports of vaccine-preventable diseases.

Disease	Dates, Number of Affected Voyages (Duration)	Diagnosis/Case Definition	No. Cases (Crew, Pax, Other)	Age, Sex, Nationality of Cases	No. Hospitalizations (HR), No. Deaths	Crew Attack Rate (of Susceptible)	Place of Occurrence/Ship Type	Vaccination/Infection History	Source/Risk Factor	Control Measures Described *	Reference
**Rubella**	2 May–28 June 1996, 1 (2 months)	Rubella IgM+ or IgG+ plus clinical exanthema	20 (20, 0, 0)	18–33 years (mean 22.6); all male, German	0 (0%), 0	6% crew (57%)	Unkn. ^‡^. (ashore)/Military (German Navy ship)	A total of 9% (*n* = 27) had documentation of vaccination status, 81% (*n* = 242) had not been vaccinated or did not know status, 36% (*n* = 98) of crew indicated they had rubella in the past—of which three became cases; **Susceptibility rate**: 12% (35/292) including infected	Crew; Hypothesized source: off ship pre-employment/Highest risk of infection: sleeping aboard ratings deck	A, C, E	[3]
	April–June 1997, multiple (3 months)	Rash illness	7 (7, 0, 0)	Unkn., unkn. 95% not US-born	0 (0%), 0	0.8% crew (unkn)	Florida (USA) to Bahamas/Passenger (cruise)	Substantial proportion (96%, *n* = 865) of crew had no documentation of rubella vaccination or immunity	Unkn.	B, I	[34]
	30 May–2 August 1997, multiple (2 months)	Rubella IgM+ or clinical Rubella plus epi-link to confirmed case	16 (16, 0, 0)	Unkn., unkn. 85% not US born from 50 countries	0 (0%), 0	4.2% crew (64%)	Florida (USA) to Bahamas/Passenger (cruise)	75% of crew had negative or unknown vaccination history, **Susceptibility rate**: 11% (41//366) including infected	Unkn.	B, E, F, G, I	[34]
**Measles**	2 February–18 July 2008, N/A (6 months)	Confirmed or epi-linked clinical measles	155 (2, 1, 152)	Age range: 5 months-41 years (38% 20–29 years: 19% <2 years); 54% male, unkn.	13 (8,4%),0	N/A	Algeciras (Spain)—Tanger (Morocco)/Passenger (ferry) and Land (Algeciras and surrounds)	72.1% of cases not previously vaccinated	Index cases were 2 crew and 1 passenger	A, F, G, N (vaccination and Ig to susceptible contacts)	[22]
	19–26 Aug 2011, Various (8 days)	Confirmed measles	5 (5, 0, 0)	Unkn., 4 male, 1 female, Philippines: 3, Honduras: 1, Italy: 1	0 (0%), 0	0.5% (unkn.)	Spain, Malta, Italy/Passenger	Unkn.	Unkn.	A, B, C, D, E, F, G, H, I, J	[21]
	8 February–July 2014, N/A (6 months)	Epi-linked and confirmed	80 (0, 1, 79)	Median age 26 years (range: 8 months–55 years); 50 (62.5%) female, N/A	35/78 (44.9%), 0	Unkn.	Sardinia, Italy/Land	74/76 cases (97.4%) unvaccinated: 2 (2.6%) with one dose of measles vaccine	Cruise ship passenger	H, G, N (all susceptible staff invited for vaccination)	[28]
	20 February–10 March 2014, 1 (7 days)	Clinical, epi- linked and confirmed	29 (23, 6, 0)	1–42 years (median 26); 21/27 (78%) male, Asia (71%), Europe (21%), S. America and Caribbean (7%), Africa (0.5%) Median age 19 years (range: 0–39 years); 17 (53%) female, Italy: 6, India: 5, Philippines: 3, Honduras: 2, Austria: 1, Brazil: 1, Indonesia: 1	10/27 (37%), 0	2.4% (unkn.)	Mediterranean Sea (Italy, France, Spain)/Passenger (cruise)	Vaccination status of 24 cases: unkn. (*n* = 12), unvaccinated (*n* = 9), vaccinated with one to two doses (*n* = 2, *n* = 1). Of crew, 142 recalled vaccination and 108 history of measles = 150/968	Index case: crew member(s): infected during cruise, possibly at one of stops	A, B, E, F, G, H, I, J, K, O	[3,31]
	27 February–May 2014, N/A (3 months)	Clinical, epi- linked and confirmed	32 (0, 1, 31)	Unkn.	Unkn.	Unkn.	Brindisi province, Puglia Region, Italy/Land	Unknown; 1/32 cases vaccinated with one dose of MMR	Cruise ship passenger	H, N (MMR vaccination for 2 cases on land as PEP and of close contacts)	[33]
**Varicella**	1998, unkn. (unkn.)	Clinical varicella case	3 (3, 0, 0)	Unkn., Most foreign-borne, many from tropical climates	Unkn.	<1% (unkn.)	New York Harbor/Passenger (cruise)	Susceptibility rate including infected 13%.	Unkn.	A, C, I	[32]
	28 January–15 April 2006 Multiple (7 days), (11 weeks)	Rash illness	16 (16, 0, 0)	Index cases 23 and 35 years, unkn., Ukraine and Philippines. Secondary cases: unkn., unkn., majority from tropical countries.	0 (0%), 0	1.3% (unkn.)	Florida (US) to Caribbean./Passenger (cruise)	Three crew members (<1%) had proof of immunity (vaccine record) to measles and rubella	All index cases were crew members; close crew interactions outside of work as risk factor	A, B, E, F, G, H, I	[1]
	February–November 2008 5 ships (unkn.), (10 months)	Clinical varicella case	28 (28, 0, 0)	Unkn., 70% crew from subtropical/tropical countries	0 (0%), 0	Unkn.	Mediterranean Sea/Passenger	Unkn.	Unkn.	I	[23]
	20 February–30 March 2012, 1 (5 weeks)	Clinical varicella case	3 (2, 1, 0)	4 years, male, unkn.	0 (0%), 0	<1% (unkn.)	UK, Spain, Portugal/Passenger	Unkn.	Unkn.	A, B, C, D, E, G, I	[21]
	29 September–2 October 2014, 1 (4 days)	Clinical varicella case	1 (1, 0, 0)	27 years, male, unkn.	0 (0%), 0	Unkn.	Malta, Spain/Passenger	Unkn.	Unkn.	A, B, C, D, E, G, I	[21]
	16 January–20 February 2015 Various (5 weeks)	Clinical varicella case	5 (5, 0, 0)	34 years, male, unkn., Indonesia	0 (0%), 0	Unkn.	Spain, Italy/Passenger	Unkn.	Unkn.	A, B, C, D, E, G, I	[21]
	26–30 December 2015, N/A (1 week)	Varicella pneumonia	1 (1, 0, 0)	50 years, male, Indian	1/1 (100%), 1 (cause of death varicella pneumonia)	1/24 (unkn.)	Puerto Rico/Cargo	Unkn.	Unkn.	A, B, E, I	[25]
	12–21 September 2016		5 (5, 0, 0)	Unkn.	0 (0%), 0	Unkn.	Belgium/Cargo ship	Unkn.	New crew members from the Philippines arrived (22/08/2016) prior to the outbreak or by contacting people in the port	A, B, C, E, I	[21]
	February 2018, 1	Clinically diagnosed varicella	1 (1, 0, 0)	Unkn., male, Philippines	0 (0%), 0	Unkn.	Spain/Container ship	Unkn.	Unkn.	A, B, I, E	[21]
**Hepatitis A**	1 September–30 November/2008, 3 (3 months)	Clinically compatible case with IgM anti-HAV, disease onset 1 Sept–30 Nov and travel history to Egypt 2–6 weeks prior to symptom onset	10 (0, 10, 0)	Median 41 years (range 23–59); male to female ratio 3:7, N/A	1 (10%), 0	Unkn.	Nile river, Belgium/Passenger (river cruise	No case vaccinated against hepatitis A	Continuing common source, most likely linked to river cruise	B	[37]
	September–November 2008, 6 (5–14 days), (10 weeks)	Hepatitis A cases (symptoms + lab confirmation of acute infection) with onset from 1 Sept 2008 and travel to Egypt 15–50 days prior to symptom onset	34 (0, 34, 0)	Mean age 40.1 years (range 11–69); 59% female, N/A	20 (59%), 0	Unkn.	Nile river, Germany/Passenger (river cruise	No case vaccinated against hepatitis A	Continuing common source, most likely linked to river cruise	B	[30]
	13 September–28 October 2009, 5 (9 weeks)	Person with IgM anti-HAV who had stayed in Egypt 2–6 weeks prior to symptom onset	26 (0, 26, 0)	Mean age 32.8 years (range 10–65); 50% female, N/A	17 (65%), 0	Unkn.	Nile river, France/Passenger (river cruise	No case vaccinated against hepatitis A	Continuing common source, most likely linked to river cruise	B	[36]
	January 2011, 1 (unkn.)	Subgenotype IB imported case	1 (1, 0, 0)	28 years, male, Polish	0, 0	Unkn.	Argentina (USA–South American Pacific–Atlantic Coast)/Passenger (river cruise	Self-report of one dose hepatitis A vaccine in previous year	Crew index case; Possible risk factor: consumption of shellfish on land in Mexico	O	[27]
**Mumps**	15 June–13 August 1992, 1 (unkn.) (2 months)	Clinical mumps	9 (9, 0, 0)	Mean 24 years (Range: 18–35); unkn., unkn.	2 (22%), 0	4% (unkn.)	Western Pacific, route: Japan to Hawaii (US)/Military (“US Reuben James” Navy ship)	Three cases had mumps or MMR vaccine documented	Unkn.	O	[35]
**Diphtheria**	1997, 1 (12 days)	Confirmed *Corynebacterium diphtheriae*	1 (0, 1, 0)	72 years, female, unkn.	1 (100%), 0	N/A	Baltic sea/Passenger (cruise)	Unkn.	Travelling in former USSR/Baltic	C, N (Diptheria antitoxin and antibiotics to case and close family, contacts received antibiotic prophylaxis and low dose diphtheria vaccine boosters)	[24]
**Meningitis**	May 2003, 1 (unkn.)	Clinical meningococcal meningitis	1 (1, 0, 0)	24 years, male, unkn.	0, 0	Unkn.	Mid Atlantic ocean/Military (aircraft carrier)	Patient had received meningococcal vaccine three years previously.	Unkn.	A, C, H, N (Chemoprophylaxis to close contacts from the ship)	[26]
	October 2012, 1 (unkn.), (<1 month)	Confirmed *Neisseria meningitides*	4 (4, 0, 0)	Unkn., unkn., originated from three different continents	4 (100%), 1 (25%)	Unkn.	Italian coast/Passenger (cruise)	Unkn.	All cases worked in ship kitchen	B, N (Chemoprophylaxis to all passengers and crew on ship)	[29]
	October 2017, 1	Clinical meningitis	1 (1, 0, 0)	Unkn., female, unkn.	1, unkn.	Unkn.	Greece/Passenger (cruise)	Unkn.	Unkn.	A, B, D, I, O	[21]

HR: Hospitalization rate; CFR: Case fatality rate; * Control measures: A = isolation, B = report to competent authority, C = treatment, D = cleaning/disinfection, E = case finding/active surveillance in crew, F = case finding/active surveillance in passengers or others, G = risk communication, H = contact tracing, I = crew vaccination, J = passenger vaccination, K = quarantine, L = immunoglobulin (Ig) for crew, M = immunoglobulin (Ig) for passengers, N = other post-exposure prophylaxis (PEP) (describe details), O = disembarkation. ^‡^ Unkn. = unknown

**Table 2 ijerph-16-02713-t002:** Vaccine-preventable diseases identified through record review studies.

Disease	Dates (Number of Affected Voyages, Duration)	Diagnosis/Case Definition	No. Cases (Crew, Passenger, Other)	Age (Years), Sex, Nationality of Cases	Population at Risk/Crew Attack Rate (of Susceptible)	Place of Occurrence/Ship Type	Control Measures Described *	Reference
**Varicella**	2005–2009 (unkn., 5 years)	Clinical varicella case	357 (278, 0, 79)	Crew: Median age 29 (range 20–66), 80% male Three-quarters of crew cases were from Caribbean countries, Indonesia, the Philippines, or India Most cases in spring and winter	2305 maritime illness reports	Sailing in US waters/passenger	A, B, C, E, F, G, H, I, J	[40]
	November 2007–April 2008 (4 ships, 6 months)	Clinical varicella case	5 (5, 0, 0)	26–42 years, male Majority from SE Asia and Eastern European countries. Crew cases: Indonesia (2), the Philippines (1), Sri Lanka (2) Index: child passenger on cruise (1) and crew coming from home country during incubation period (3)	13, 377, 28, and 882 crew, respectively	Baltic sea/Europe/Med/USA and Caribbean/passenger (2) cargo (1) work (feeder) ship (1)	A, B, C, E, F, G, H, I, J (instructed but unclear what was carried out)	[39]
	2009–2010, 34 ships (2–25 days, 2 years)	Clinical varicella case (probable case)	187 (151, 36, 0)	Unkn. >130 countries. 26 countries for crew cases with 58% from 5 sub-tropical or tropical countries (the Philippines, Indonesia, India, Jamaica, St. Vincent and the Grenadines)	694–6300 passengers per ship; 400–2160 crew per ship. Crew attack rate (of susceptible) For crew clusters reported: (i) 29/860 (3.4%), (ii) 10/2400 (0.4%)	World-wide/passenger	A, B, C, E, F, G, H, I, J	[38]
	January 2010–December 2015, 99 outbreaks	Diagnosis of authority’s medical officer	967 (823 including 758 cruise ship crew, 144, 0)	Crew: 20–49: 394/399, 50+: 5/399, Pax: <1: 3/99, 1–9: 46/99, 10–19: 18/99, 20–49: 24/99 50+: 8/99 Male 395/479 crew 60/106 pax Female 84/479 crew 46/106 pax Indonesia (21.7%, 80/369), Philippines (17.6%, 65/369), or India (17.3%, 64/369)	Unknown	USA/63 cargo ships, 900 cruise ships	A, B, C, H, I, J, O	[41]
**Herpes zoster**	January 2010–December 2015	Diagnosis of authority’s medical officer	13 (9 including 7 cruise ship crew, 4, 0)	Unkn.	Unkn.	USA/2 cargo ships, 11 cruise ships	Unkn.	[41]
**Mumps**	January 2010–December 2014	Presumptive or final diagnoses based on the information available from healthcare facilities, health departments, medical examiners, or laboratories, when available, and review by a quarantine medical officer	35 (13, 22, 0)	Unkn.	Unkn.	Unkn.	Unkn.	[11]
**Meningitis/meningococcal disease**	January 2010–December 2014	Presumptive or final diagnoses based on the information available from healthcare facilities, health departments, medical examiners, or laboratories, when available, and review by a quarantine medical officer	25 (9, 16, 0)	Unkn.	Unkn.	Unkn.	Unkn.	[11]
**Hepatitis A**	January 2010–December 2014	Presumptive or final diagnoses based on the information available from healthcare facilities, health departments, medical examiners, or laboratories, when available, and review by a quarantine medical officer	22 (19, 3, 0)	Unkn.	Unkn.	Unkn.	Unkn.	[11]
**Measles**	January 2010–December 2014	Any probable or confirmed measles case determined to be contagious during travel. Presumptive or final diagnoses based on the information available from healthcare facilities, health departments, medical examiners, or laboratories, when available, and review by a quarantine medical officer	21 (10, 11)	Unkn.	Unkn.	Unkn.	Unkn.	[11]
**Rubella**	January 2010–December 2014	Presumptive or final diagnoses based on the information available from healthcare facilities, health departments, medical examiners, or laboratories, when available, and review by a quarantine medical officer	10 (4, 6)	Unkn.	Unkn.	Unkn.	Unkn.	[11]
**Pertussis**	January 2010–December 2014	Presumptive or final diagnoses based on the information available from healthcare facilities, health departments, medical examiners, or laboratories, when available, and review by a quarantine medical officer	9 (0, 9)	Unkn.	Unkn.	Unkn.	Unkn.	[11]

**Table 3 ijerph-16-02713-t003:** Seroprevalence studies.

Disease, Study Duration [Reference]	Study Date (Duration)	Study Sample	Serological Markers *	No. Study Subjects (Age, Sex, Nationality	Seropositivity	Seropositivity By Age	Seropositivity by Nationality	Susceptibility	Vaccination/Infection History	Risk Factors
**Hepatitis Study/1989–1991** [44]	1989–1991 (3 years)	US military personnel scheduled for deployment on 11 US Navy ships	Anti-HAV+, Anti-HBc+, Anti-HBsAg+ Anti-HDV, Anti-HCV+	2072 (male, mean 24 years; 72% white)	Anti-HAV+ 210 (10.1%); Anti-HBc+ 76 (3.7 %); Anti-HCV+ 9 (0.4%)	Increasing with age: HAV+ 7.8% in 18–24 years; 28.7% in >34 years Anti-HBc+: 2.2% in 18–24 yo; 7.2% in 25–34 years; 8.3% in >34 years	For country of birth: Anti-HAV+ 8% for USA and 39.3% for foreign; HBV+ for 3.2% USA, 10.7% for foreign	HAV: 90%	28 subjects reported a history of acute hepatitis (50% were anti HAV+ compared to 9.6% without a history)	Anti-HAV positivity associated with age, non-white racial/ethnic group, born outside US and prior Caribbean deployment for <1 year. Anti-HBV with black and Philippino race, foreign birth, a history of STD and South Pacific/Indian Ocean deployment for <12 months, and S Pacific or Med duty for >1 year
**Hepatitis Study/1993** [43]	April–December 1993 (9 months)	Seamen attending five clinics performing mandatory health examinations of seamen in Denmark	Anti-HAV+, Anti-HBc+, Anti-HBsAg+, Anti-HCV+	515 (86% male; 94% Scandinavian)	Anti-HAV+ 79 (15.3%); Anti-HBc+ 47 (9.1%); Anti-HCV+ 6 (1.2%)	Increasing with age: HAV+ 0.3% in <40; 71% in 60–69 years. HBV+: 2.7% in <40; 35.7% in >60	For origin: Anti-HAV+ 12.8% for Scandinavian and 52.4% for foreign; HBV+ 7.9% for Scandinavian, 28.5% for foreign	HAV: 85%	1 HAV+ case previously had HAV vaccination	Anti-HAV and HBV positivity associated with age. HAV seroprevalence highest among those who sailed in international trade (outside USA and Europe)
**Hepatitis Study/1998** [45]	February–July 1998 (6 months)	Personnel from a Greek warship	Anti-HAV+, Anti-HBsAg+ Anti-HBc+, Anti-HBs+, Anti-HCV+	263; (male, mean age 24.4 years)	Anti-HAV+ 0 (0%); Anti-HBsAg+ 3 (1.1 %); Anti-HBc+, 4 (1.5%) Anti-HBs+ 45 (17.1%); Anti-HCV+ 1 (0.4%)	Unkn.	Unkn.	HAV: 100%	23 subjects reported vaccination against Hepatitis B (three doses by 14 subjects)	Unkn.
**Varicella Study/2008** [42]	1–23 December 2008 (3 weeks)	Cruise ship crew members undergoing pre-employment medical exam in Mumbai and Goa	VZV IgM+ and IgG +	121 (male, 21–42 years, 100% Indian)	100 (82.7%) IgG+ 0 (0%) IgM+	Unkn.	Unkn.	16.5% susceptibility	60% IgG pos. crew could remember disease or vaccine history	Unkn.

* anti-HAV: antibodies to hepatitis A virus, anti-HBc: antibodies to hepatitis B core antigen, anti-HCV: antibodies to hepatitis C virus, Anti-HBsAg: antibodies to hepatitis B surface antigen, VZV: Varicella Zoster Virus, IgM: immunoglobulin M, IgG: immunoglobulin G, Unkn.: unknown, STD: Sexually Transmitted Disease.

**Table 4 ijerph-16-02713-t004:** Characteristics of cases from reported outbreaks, case reports, and record reviews (results from seroprevalence studies not included).

Disease	No. Clusters/Outbreaks > 1 Cases	No. Crew Cases		No. Pax Cases		Total Crew, Pax/Total Cases *n* = 1795 (%)	No. Other Cases	Male/Female	Origin of Cases	Ship Type
		Out. Rep. ^‡^	Rev.Rec. ^∞^	Out. Rep.	Rev. Rec					
Measles	3	30	10	7	11	40, 18/58 (3, 2)	262 *	25/7	Asia, Europe, S. America, Caribbean, Africa, Italy, India, Philippines, Honduras, Austria, Brazil, Indonesia, unkn.	1 ferry, 2 cruise
Rubella	3	43	4	-	6	47, 6/53 (3, 0)	0	20/-	German, unkn.	2 cruise, 1 military
Hepatitis A	3	-	19	70	3	19, 73/92 (5, 1)	0	31/40	Polish, unkn.	4 river cruise ships
Meningococcal meningitis	1	4	9	-	16	13, 16/ 29 (1, 6)	0	1/1	unkn.	2 cruise, 1 military
Mumps	1	9	13	-	22	22, 22/44 (2, 5)	0	-/-	unkn.	1 military
Pertussis	unkn ^√^.	-	-	-	9	0, 9/9 (0, 5)	0	-/-	unkn.	-
Herpes zoster	unkn.	-	9	-	4	9, 4/13 (0, 7)	0	-/-	unkn.	11 cruise, 2 cargo
Varicella	104	59	1257	1	180	1316, 181/1497 (83, 4)	79 ^†^	687/130	Indonesia, India, Philippines, SE Asia, Eastern European countries, Sri Lanka, Caribbean countries, unkn.	909 cruise, 69 cargo
**Total**	**115**	**145**	**1321**	**78**	**251**	**223, 1572/1795**	**341**	**764/179**		**1006**

* For example, secondary cases on land, ^†^ either passenger cases or their contacts, ^‡^ Outbreak Report (Out.Rep.), ^∞^ Review Record (Rev.Rec.), ^√^ Unknown (unkn.).

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
