# Peer review of "A Systematic Review for Vaccine-Preventable Diseases on Ships: Evidence for Cross-Border Transmission and for Pre-Employment Immunization Need"

_ijerph, 2019, doi:10.3390/ijerph16152713_

Round 1

Reviewer 1 Report

Dear Authors,

I have read with great interest the manuscript titled “A systematic review for vaccine preventable diseases on ships: evidence for cross-border transmission and for pre-employment immunization need”. In general this is a well-executed systematic review.

In my opinion the manuscript requires a limitation paragraph in the discussion section. This paragraph should contain the points of the PRISMA methodology that have not been met and how the authors have resolved this issue. The quality of the articles, in this case, can be evaluated through the STROBE checklist (STrengthening the Reporting of OBservational studies in Epidemiology) based on 22 points and how it could affect the results obtained. Another question that arises is whether all the outbreaks of VPD on ships have been reported through scientific literature. I do not think so, I believe there are cases that are not included in this review. These outbreaks could correspond to ships from countries without much scientific tradition. How do they influence the results? Would the conclusions of the article be affected? Personally, I think the review only shows a small part of a problem that is surely much bigger.

Congratulations for the work.

Author Response

We would like to thank the reviewer for giving us the opportunity to clarify additional points in the limitations of this study. We have modified the discussion as follows considering the   comments and suggestions:

The objectives of this review were achieved by collecting evidence mainly from published data. It is possible that most cases and outbreaks are not published in the scientific literature. The majority of cases were varicella cases collected by the US reporting system. Data collected in this review are mainly from US and Europe. In Europe, even if there is an information system available to MS to record cases and public health measures information, it is not systematically used in all EU MS. Moreover, it is possible that cases or outbreaks have been reported to other non-EU countries, but those were not considered in our study. It is unknown how many cases and outbreaks are not reported to competent authorities.

Reviewer 2 Report

This systematic review for vaccine-preventable diseases on ships had collected evidence recent two decades for cross-border transmission and for pre-employment immunization need. Generally, the whole manuscript is well prepared and is good to update our current knowledge for this field.

Author Response

We thank the reviewer for the review and comments. 

Reviewer 3 Report

The authors provided a well-written review for vaccine preventable diseases on ships, which is benefited for controlling cross-border transmission. Advises as follows:

Could the authors provide a map about the outbreaks or case reports spots?

It is suggested to describe the outbreaks, cases etc. in chronological order basically.

Author Response

We thank the reviewer for this comment, we have ordered the tables chronologically as suggested.